# Cancer Metastasis Detection Through Multiple Spatial Context Network

Wutong Zhang[1], Chuang Zhu[1], Jun Liu[1], Ying Wang[2], and Mulan Jin[2]

[1] Center for Data Science, Beijing University of Posts and Telecommunications, Beijing, China
zhangwutong,czhu,liujun@bupt.edu.cn
[2] Capital Medical University, Beijing, China
wangying_ng_blk@126.com, kinmokuran@163.com

**Abstract.** Breast cancer is one of the leading causes of death by cancer in women, and it often requires accurate detection of metastasis in lymph nodes through Whole-slide Images (WSIs). At present, there are many algorithms of cancer metastasis detection based on CNN, which are generally patch-level models, aiming for increasing the sensitivity, speed, and consistency of metastasis detection. However, most of these algorithms use patch as an independent individual to train, which leads to the neglect of much important spatial context information in WSI. In this paper, we propose a multiple spatial context network (MSC-Net) which considers the spatial correlations between neighboring patches through fusing the spatial information probability maps obtained from the two novel networks we propose, the self-surround spatial context stacked network (SSC-Net) and the center-surround spatial context shared network (CSC-Net). The SSC-Net is a deep mining of continuous information between patches, while CSC-Net strengthens the influence of the neighborhood information to the central patch. Furthermore, for saving memory overhead and reducing computational complexity, we propose a framework which can quickly scan the WSI through the mechanism of the patch feature sharing. We demonstrate evaluations on the camelyon16 dataset and compare with the state-of-the-art trackers. Our method provides a superior result.

**Keywords:** Deep Learning · Spatial Context Relation · Cancer Metastasis Detection.

## 1 Introduction

Worldwide, there are about 2.1 million newly diagnosed female breast cancer cases in 2018, accounting for almost 1 in 4 cancer cases among women [2]. Actually, more than 90% of women diagnosed with breast cancer at the early-stage survive their disease for at least 5 years. Therefore, the early cancer diagnosis and treatment play a crucial role in improving patients survival rate. Specifically, during the diagnosis procedure, specialists evaluate both overall and local tissue organization via Whole-slide Images (WSIs). However, manually detecting

tumor cells within extremely large WSIs can be tedious and time-consuming. Furthermore, it has been shown that there is limited inter-observer consensus in interpreting breast biopsy specimens [8]. Because of this, the development of automatic detection and diagnosis tools is challenging but also essential for the field. And it is the hotspot to develop algorithms to detect cancer metastasis in lymph node images using computer assisted detection.

For decades, with the advent of convolutional neural networks (CNNs) and their excellent performance for natural image classification [5,9], there is a growing trend to adapt CNN in computer assisted detection of lymph node metastasis in WSIs [4, 10]. Usually, because of the extremely large size of WSIs, most of the studies first extracted small patches (e.g. 256 * 256 pixels) from WSIs, and trained a deep CNN to classify these small patches into normal or tumor regions [10, 13]. However, these algorithms train each patch independently, which leads to a serious problem that the loss of spatial context information and dependency in WSI. Therefore, during inference time, the predictions over neighboring patches may be inconsistent, and the patch level probability map may contain isolated outliers. Actually, according to diagnostic experience, when a patch is in the tumor region, its neighboring patches also have a high probability to be labeled as tumor, since they are co-located in neighboring regions [6].

In order to capture spatial neighborhood information, Bin Kong [6] proposed Spatio-Net that uses 2D-LSTM layers. But the 2D-LSTM may causes a heavy computational overhead burden and makes the training process extremely slow. And Yi Li [7] propose a neural conditional random field (NCRF) deep learning framework. However, the spatial dependencies in NCRF which is a post-processing method are always suboptimal because complex configurations of patch.

In this work, we first propose SSC-Net that can capture continuous spatial information more comprehensively on the internal structure of a single patch. Then, we develop the CSC-Net to mine discrete spatial information with fixed directions around one center patch, which is complementary to the SSC-Net. Finally, we fuse the spatial information probability maps obtained from the above two networks and use sliding windows to get the whole WSI prediction results.In addition, for alleviating the memory consumption problem when sliding windows on the whole WSI, we propose a fast scanning framework by asynchronous sample pre-fetching and neighborhood feature sharing.

## 2   Methodology

### 2.1   Overview

The framework is divided into two parts as Fig.1. The first part is feature extraction using CNN. It's noting that unlike previous deep neural network methods that treat each small image patch independently, the proposed framework combines each image patch and its neighbors together for consideration. The second part uses two different components, SSC-Net and CSC-Net, to obtain continuous and discrete spatial context dependent information separately, which can

**Fig. 1.** The overall scheme of our proposed framework. First, the WSIs are divided into many small patches. Second, Each patch and its eight neighbors are fed into CNN feature extractor. Then the transform layer sends the features to the two components in the multi-spatial layer. After a series of spatial feature extraction, we fuse the results of the two components, resulting in a probability map, and finally further processed to locate the metastases.

effectively improve the accuracy of patch level and the generalization ability of the model. Then, The output of the above components is integrated for final classification.

During the training phase, we load a grid of patches, e.g.3x3, only the predicted probability of the center patch is retained for easy implementation. In the testing phase, we perform inference over patches in a sliding window across the slide, generating a tumor probability heatmap, but it is heavy computational overhead. Therefore, we propose a fast scanning framework to optimize the conventional sliding window structure.

## 2.2    Feature Extractor With CNN

Unlike Hand-crafted features [14,15], CNN feature extractor preserve the inputs neighborhood relations and spatial locality in their latent higher-level feature representations.Therefore, using CNN as feature extractor can not only retain the important spatial relevance of images, but also greatly reduce the dimension of features, which makes it easier to capture spatial context information. In our framework, we employ two ResNet architectures [5], ResNet-18 and ResNet-34 that have proven to be powerful in image classification task to extract comprehensive feature representation of pathological image. After the transform layer, we will get a grid of patch feature.

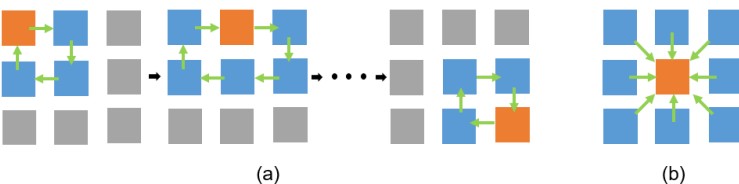

**Fig. 2.** The structure of SSC-Net and CSC-Net.(a)SSC-Net: The blue node is the current neighborhood patch, the orange node is the current center patch, and the LSTM follows the direction of the arrow from the center patch to bypass all the neighborhood patches and finally return to the center patch.(b)CSC-Net: For the center patch, multi-directional parallel LSTMs are used to perform discrete fixed-direction neighborhood information mining based on the direction of the arrow.

### 2.3   Self-Surround Spatial Context Stacked Network

In order to capture the connectivity information of the space more comprehensively in grid of patch feature, we designed a separate closed-loop LSTM for each individual patch feature as represented by the circle in Fig.2(a). Through such a closed-loop LSTM structure, continuous spatial context information around each patch can be captured and the continuous dependence of neighborhood patch is also preserved. Then, after obtaining N spatial neighborhood information, we will get a new feature map which will be fed into the next same stacked layer which can obtain spatial context information at a higher semantic level. The SSC-Net formula referring to the standard LSTM [12] can be simplified as:

$$O_{t+1}^d, (m_{t+1}^d, c_{t+1}^d) = LSTM(I_t^d, (m_t^d, c_t^d)) \tag{1}$$

where $I_t^d$ is the current input; $O_{t+1}^d$ is the final output;$(m_t^d, c_t^d)$are the long and short memory state; t is used to control the order of input blocks of LSTM. For example, if the current center position is (i, j), then the sequence of t is a set of rounds from the center around the center and finally returned to the center position; d is the number of layers stacked.

This model can capture the surrounding context information of each patch through the closed-loop LSTM connection, and then make the relationship between each patch more tight through the stacking of the same layer. In this way, we can obtain a feature that incorporates neighborhood information that is logically linked in a certain order.

### 2.4   Center-Surround Spatial Context Shared Network

The SSC-Net was a deep mining of continuous logical sequence of spatial context information, while CSC-Net obtains discrete spatial context information in the eight different fixed directions and strengthens the study of the neighborhood information of the central patch, like Fig.2(b). Because of traditional 2D-LSTM [3] can only take into account the neighborhood information on the left and

the top of the center patch, it is very incomplete. So we have adopted a novel network based on Multi-directional parallel LSTMs, which can process the full spatial context of each patch in such a WSI through eight sweeps over all patch by eight different LSTMs. The formula is denoted as follows:

$$O_{t+1}^1, (m_{t+1}^1, c_{t+1}^1) = LSTM(I_t^1, (m_t^1, c_t^1))$$
$$O_{t+1}^2, (m_{t+1}^2, c_{t+1}^2) = LSTM(I_t^2, (m_t^2, c_t^2))$$
$$\vdots$$
$$O_{t+1}^N, (m_{t+1}^N, c_{t+1}^N) = LSTM(I_t^N, (m_t^N, c_t^N))$$

$$(2)$$

where $I_t^d$ is the current input; $O_{t+1}^d$ is the final output;$(m_t^d, c_t^d)$are the long and short memory state; t is used to control the order of input blocks of LSTM. N represents the number of adjacent patches(N= 8 in our case).

## 2.5  Multiple Spatial Context Information Integration Network

After passing through the above two components in parallel, the grid of patch feature which contains spatial context information enters the fully connected layer to classify, and outputs two grid of patches classification result. The grid of patches classification result obtained by the SSC-Net emphasizes the continuous spatial dependence of each patch and the patch around itself while the result of SSC-Net further emphasizes the influence of the discrete fixed direction on the spatial structure of the center patch. These two components form a certain degree of spatial domain context information complementarity. Therefore, we combine the spatial information probability maps obtained from the above two networks to obtain the final prediction results.

## 2.6  A Fast WSI Scanning Framework

**Asynchronous Sample Prefetching** During the training phase, the heavy I/O bottleneck always exists, i.e., the GPU is often idle while waiting for fetching batched training data. To resolve this problem, we adopt an asynchronous sample prefetching mechanism by using multiple producer processes of CPU to prepare the training samples while one consumer process for GPU to consume the training data. This strategy can keep GPU running all the time and boostat least 10 times acceleration in the training stage.

**Neighborhood Feature Sharing** In the testing phase, we perform inference over patches in a sliding window across the slide, generating a tumor probability heatmap, but it is heavy computational overhead. Therefor, we adopt feature sharing method to avoid repetitive computation and improve scanning efficiency, as shown in the Fig.3. The merit of using neighborhood feature sharing architecture. It can speed up the inference by sharing computations in the overlapping regions (blue patch).

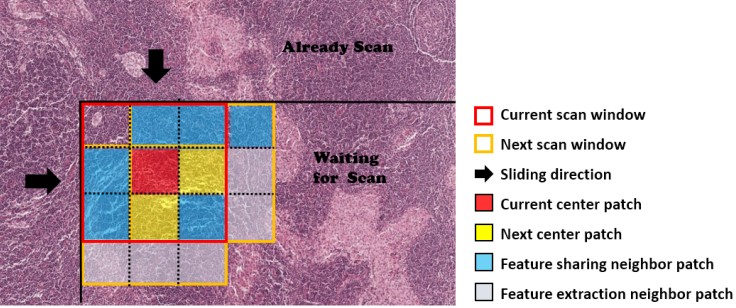

**Fig. 3.** A Fast WSI Scanning Framework. When predicting the center block, the feature maps of the eight neighborhoods need to be calculated simultaneously. Then, when predicting the next adjacent center block (right side or bottom side), it is possible to calculate only the feature map of the newly read patch (gray block), and the block that has been used last time (blue block) can be used without calculation.

## 3    Experments

In this section, extensive experiments were conducted on the CAMELYON16 [1] dataset to evaluate the proposed model for cancer metastasis detection in WSIs. This dataset includes 160normal and 110 tumor WSIs for training, 81 normal and 49 tumor WSIs for testing. We conducted all the experiments on $40\times$ magnification.First, We employ the simple OTSU algorithm [11] to determine the adaptive threshold and filter out most of the white background. Then, We randomly sampled 200,000 $768 \times 768$ patches from the non-tumor non-background regions of the tumor slides and the non-background regions of the normal slides as negative samples. In order to probe the efficacy of our method, we first evaluate our model under different configurations. We tried to use different CNN feature extractors. And Experiments show that using a ResNet18 network is enough to extract the appropriate features while saving memory. Our baseline is directly using the ResNet18 network. We also compared our method with several state-of-the-art methods using accuracy as evaluation indicator.

As shown in Table 1, on the full datasetall of the model proposed in this paper SSC-Net,CSC-Net and Multi-Net have a higher accuracy. And as expected, Multi-Net has the highest accuracy, which is 6.72% higher than baseline, in the case of guaranteeing high FROC. At the same time, it is worth noting that our model works better on a small number of data sets than other models because of the combination of domain information. For most of the depth models, with the increase of the complexity of the model, it may make the model over-fitting in a small number of data sets serious, so that the performance on a small number of data sets is not as good as the simple model.

Fig.4 is the curves of the training process on 10% of dataset. As analyzed above, show that our model still has smooth training curves with small amount of data, contrast the fluctuation of baseline. Therefore, our model has a natural

**Table 1.** Quantitative comparisons

| Model | ACC(10%Data) | ACC(100%Data) | Ave.FROC | STD |
|---|---|---|---|---|
| Baseline | 88.52% | 92.42% | 0.4301 | 0.026 |
| ResNet34 | 88.62% | 92.76% | 0.5241 | 0.023 |
| ResNet50 | 88.68% | 93.57% | 0.5249 | 0.019 |
| NCRF | 91.15% | 92.96% | **0.8138** | 0.010 |
| SSC-Net | 92.07% | 92.59% | 0.7825 | 0.011 |
| CSC-Net | 94.13% | 97.54% | 0.7526 | 0.012 |
| MSC-Net | **95.24%** | **98.43%** | **0.8078** | 0.010 |

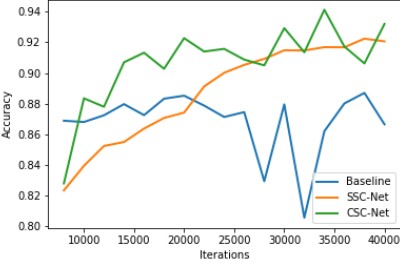 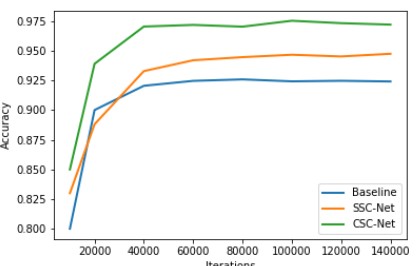

**Fig. 4.** Accuracy on 10% of dataset          **Fig. 5.** Accuracy on full dataset

strong generalization ability when the amount of data is small, due to the use of stack LSTM. Therefore, the training efficiency of the model can be greatly improved.

## 4    Conclusion

In this paper, we propose a novel multiple spatial context network, which is composed of SSC-Net and CSC-Net, and through integrate neighborhood and background features improve the detection of metastasis in WSIs. The SSC-Net and CSC-Net which are based on the LSTM. A standard LSTM allows to easily memorize the context information for long periods of time in sequence data. In images, this temporal dependency learning is converted to the spatial domain which is significance for us to obtain continuous spatial dependencies. Therefor, the SSC-Net and CSC-Net generalize standard LSTM by providing recurrent connections along with all spatial dependence present in the data. Moreover, we propose a fast scanning framework by asynchronous sample prefetching and neighborhood feature sharing to alleviate the memory consumption problem when sliding windows on the whole WSI. We demonstrate that the proposed method achieved superior performance compared to other state-of-the-art methods on the Camelyon 2016 Grand Challenge dataset and even surpassed human performance. Furthermore, the proposed fast WSI scanning framework matched the speed requirements of clinical practice, where the framework can

process whole-slide image within a very short time. We expect that our multiple spatial context network is useful to boost performance in a variety of medical image analytical challenges.

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
