# OpenReview forum: "Cancer Metastasis Detection Through Multiple Spatial Context Network"
_MICCAI.org/2019/Workshop/COMPAY — Submitted to COMPAY 2019_

### Official Review · AnonReviewer2 · 2019-07-30
**Aggregating features from neighboring patches to detect cancer metastasis**

**Rating:** 6
**Confidence:** 4

**Review:**

The authors present an approach to detect cancer metastasis in publicly available CAMELYON16 challenge data set. The authors aggregate features learnt from neighbouring patches with the help of a combination of LSTM networks. The paper proposes an asynchronous sample prefetching method to improve training time but do not provide any details into it which is understandable given the limitations of the paper. But there is still room for a few details. The results are compared against different versions of ResNet architecture and the authors completely ignore the results published in CAMELYON16 paper. FROC results are provided which do not seem to improve compared to the results published in CAMELYON16 paper. No results are provided for AUC as proposed in the CAMELYON16 paper.
Overall there are many grammatical mistakes and typos in the paper which needs improvement.

---

### Official Review · AnonReviewer1 · 2019-07-31
**Approach combining two LSTM models to incorporate spatial context into tissue type classification**

**Rating:** 5
**Confidence:** 4

**Review:**

The paper proposes using between patch spatial context to increase CNN accuracy for patch based classification (applied to Camelyon 2016). Context is obvious important in such applications. The method itself isn't what you would call the most obvious approach. It uses a pair of LSTM networks (presumably Long Short Term Memory networks although LSTM is never defined - It should be and referenced) applied to the output of a retrained feature network (I'm assuming pre-trained here, training never specified explicitly). To me this is an odd approach to use a 1D network for a problem that is explicitly 2D. However, a significant improvement on the baseline is shown. The trouble is the baseline is very far from state of the art. They claim better than state of the art performance on this data set, but it's hard to say as they report results as Accuracy (ACC) rather than Area under curve (AUC) as is standard for this data set. It's hard to work out if the experimental validation methodology is consistent with other authours so I'll say the Jury is out on this claim (state of art is currently >.99 AUC!). The paper also have quite a lot of English language and ambiguity issues, for example:

Abstract "Based on CNN[s]", "is a deep mining", "state of the art trackers [what is a tracker???]",
P2: "is the hotspot to develop" [???]
P3: "Then, The.." [capitalisation]
Various: Therefor -> therefore
P6: datasetall ->dataset all

[There are others, it needs proof reading really]

Also:

P2: "For decades " - you say decades and then reference papers from last 5 years. It's true CNNs have technically been around for about 3 or 4 decades, but really their widespread use is in the last decade (or two at a push).

P2: "2D LSTM .. high computational burden in training" - why does this matter if it is fast at evaluation time? (is it?). Was it impractical to try? Your complete lack of comparative evaluation of alternative context approaches weakens the paper. Could a second 2D CNN have been used as a context network?

P5: The pre-fetching method described in 2.6 is pretty standard for large images. You could describe this with one sentence and save space.

---

### Official Review · AnonReviewer3 · 2019-08-22
**Review Cancer Metastasis Detection Through Multiple Spatial Context Network**

**Rating:** 3
**Confidence:** 4

**Review:**

This paper aims at better exploiting spatial context in whole-slide images for cancer metastasis detection, using a combination of two networks encoding spatial information. While the idea is interesting, the implementation and evaluation lack details to fully assess the work.

My main concerns are:
* The evaluation protocol is not clearly defined (no validation set ?), and no details are given with respect to parameter values and how they were tuned. If the newly proposed methods were optimized on the test set, their results might be too optimistic. Influential parameters of SSC-Net, CSC-Net, and MSC-Net are not well described and impact of their values are not studied. For example, page 3 "During the training phase, we load a grid of patches, e.g.3x3". It is the actual value used in the Section Experiments ? How was it tuned ?
* The literature on networks encoding spatial dependencies is not summed up satisfactorily in the introduction.
* There are several Resnet architectures but author's choice lacks justification and consistency: the text (Section 2 page 3) mentions Resnet-18 and Resnet-34, but in Table 1 (page 7) Resnet-34 and Resnet-50 are mentioned.
* Many writing errors (Experments -> Experiments ; Therefor -> Therefore; missing spaces between "boostat", "160normal",...). In general, the writing style could be improved.